# Geographic variation and factors associated with under-five mortality in Ethiopia. A spatial and multilevel analysis of Ethiopian mini demographic and health survey 2019

**Zemenu Tadesse Tessema**[1,2]*, **Tsion Mulat Tebeje**[3], **Lewi Goytom Gebrehewet**[2]

**1** Department of Epidemiology and Preventive Medicine, School of Public Health and Preventive Medicine, Monash University, Melbourne, Australia, **2** Department of Epidemiology and Biostatistics, Institute of Public Health, College of Medicine and Health Sciences, University of Gondar, Gondar, Ethiopia, **3** Epidemiology and Biostatistics Unit, School of Public Health, Dilla University, Dilla, Ethiopia

* zemenut1979@gmail.com

**Data Availability Statement:** Third party data was obtained for this study from The DHS Program. Data may be requested from The DHS Program

## Abstract

### Background

The distribution of under-five mortality (U5M) worldwide is uneven and the burden is higher in Sub-Saharan African countries, which account for more than 53% of the global under-five mortality. In Ethiopia, though U5M decreased substantially between 1990 and 2019, it remains excessively high and unevenly distributed. Therefore, this study aimed to assess geographic variation and factors associated with under-five mortality (U5M) in Ethiopia.

### Methods

We sourced data from the most recent nationally representative 2019 Ethiopian Mini-Demographic and Health Survey for this study. A sample size of 5,695 total births was considered. Descriptive, analytical analysis and spatial analysis were conducted using STATA version 16. Both multilevel and spatial analyses were employed to ascertain the factors associated with U5M in Ethiopia.

### Results

The U5M was 5.9% with a 95% CI 5.4% to 6.6%. Based on the multivariable multilevel logistic regression model results, the following characteristics were associated with under-five mortality: family size (AOR = 0.92, 95% CI: 0.84,0.99), number of under-five children in the family (AOR = 0.17, 95% CI: 0.14, 0.21), multiple birth (AOR = 14.4, 95% CI: 8.5, 24.3), children who were breastfed for less than 6 months (AOR = 5.04, 95% CI: 3.81, 6.67), people whose main roof is palm (AOR = 0.57, 95% CI: 0.34, 0.96), under-five children who are the sixth or more child to be born (AOR = 2.46, 95% CI: 1.49, 4.06), institutional delivery (AOR = 0.57, 95% CI: 0.41, 0.81), resident of Somali and Afar region (AOR = 3.46, 95% CI: 1.58, 7.55) and (AOR = 2.54, 95% CI: 1.10, 5.85), respectively. Spatial analysis revealed that hot spot areas of under-five mortality were located in the Dire Dawa and Somali regions.

after creating an account and submitting a concept note. More access information can be found here: https://dhsprogram.com/data/Access-Instructions.cfm The authors confirm that interested researchers would be able to access these data in the same manner as the authors. The authors also confirm that they had no special access privileges that others would not have.

**Funding:** The authors received no specific funding for this work.

**Competing interests:** The authors have declared that no competing interests exist.

**Abbreviations:** AOR, Adjusted Odds Ratio; CI, Confidence Interval; CS, Cesarean Section; DHS, Demography and Health Survey; EAs, Enumeration Areas; EMDHS, Ethiopia Mini Demography and Health Survey; HHs, households; ICC, Intraclass Correlation; KR, kids record; SGD, Sustainable Development Goal; U5M, under-five mortality; U5MR, under-five mortality rate.

## Conclusion

Under-five mortality in Ethiopia is high and unacceptable when compared to the 2030 sustainable development target, which aims for 25 per 1000 live births. Breastfeeding for less than 6 months, twin births, institutional delivery and high-risk areas of under-five mortality (Somali and Dire Dawa) are modifiable risk factors. Therefore, maternal and community education on the advantages of breastfeeding and institutional delivery is highly recommended. Women who deliver twins should be given special attention. An effective strategy should be designed for intervention in under-five mortality hot spot areas such as Somali and Dire Dawa.

## Background

Under-five mortality is defined as deaths reported at ages 0 to 59 months and includes neonatal, post-neonatal, and child deaths [1]. It is a significant indicator of the socioeconomic, health and environmental conditions, and of national development and health equity and access [2].

Globally, in 1990, the number of under-five deaths was 12,494,000 (93 deaths per 1000 live births). Following an average annual decline of 3.1%, by 2019 it was significantly reduced to 5,189,000 deaths (38 deaths per 1000 live births) representing a 58% reduction [3]. The 2030 sustainable development goals (SDG) aims to end the preventable deaths of newborns and children under 5 years of age, with all countries aiming to reduce under-5 mortality to at least 25 per 1000 live births or lower [4].

The distribution of U5MR worldwide is uneven and the global burden of U5MR is concentrated in two regions, Sub-Saharan Africa and Central and Southern Asia, which accounts for more than 80% of U5MR [5]. The burden is even higher in Sub-Saharan African countries with 74 deaths per 1000 live births, which accounts for more than 53% of the global under-five mortality [6].

In Ethiopia, U5MR has substantially decreased from 200 per 1000 live births in 1990 to 49 per 1000 live births in 2020 but is not close to Sustainable Development Goal [3]. Additionally, different studies conducted in Ethiopia revealed several socioeconomic, demographic, and geographic or spatial variation. For example, Afar, Somali, and Benishangul Gumuz are the three high-risk regions of Ethiopia for U5MR [7].

Different studies examining U5M previously showed that source of water [8], multiple births [6, 7], region [9, 10], age and sex of household head [1], maternal education status [11, 12], place of delivery [9, 10, 13], maternal age [14–16], sex of the child [8, 13, 16], residence [17, 18], family size [9, 15], number of children under five [18, 19], birth order [9, 15], and breastfeeding status [18, 20] have a significant association with under-five mortality.

Although U5M has been declining it remains high and takes the lives of many children, no research on U5M has been made until recently, with the Ethiopian mini–Demographic Health Survey (EMDHS) 2019. Research is required to inform policies and interventions for the prevention of U5M with the efficient allocation of scarce resources to priority areas based. This study aimed to assess the geographic variation and factors associated with under-five mortality in Ethiopia. The results of our study will assist in achieving the SDG, by providing an understanding of the current burden of U5M in Ethiopia, and its determinants, in addition to identifying areas and factors with the highest burden of U5M in Ethiopia.

## Methods

### Data source

Our source was the 2019 EMDHS, the second mini demographic health survey (DHS) conducted in Ethiopia (a land-locked country located in the Horn of Africa that lies between the 30N and 150N Latitude or 330E and 480E Longitude) [21]. Data collection was conducted from March 21, 2019, to June 28, 2019, the nine regions (Tigray, Afar, Amhara, Oromia, Somali, Benishangul Gumuz, Southern nation nationalities and People region (SNNPR), Harari, and Gambella) and two administrative cities (Addis Ababa and Dire Dawa). The study design was a population-based cross-sectional study. A frame of all census Enumeration areas (EAs) was used as a sampling frame for the 2019 EMDHS. 149,093 EAs were created which cover an average of 131 Households (HHS). A two-stage stratified cluster sampling technique was employed and each region was stratified into urban and rural areas, yielding 21 sampling strata were selected independently in each stratum. In the first stage, 305 clusters (93 urban and 212 rural) were selected with probability proportional to EAs size and with independent selection in each sampling stratum. In the second stage, a fixed number of 30 households per cluster was selected. Finally, women aged 15–49 in 9,150 (2,790 urban and 6,360 rural) households from 305 clusters were selected. The whole procedure of sampling is found in the full 2019 EMDHS report [21].

### Study variables

The outcome variable was under-five mortality status, which was categorized as (child alive: Yes = 0 and No = 1). The age was recorded in months. The community-level predictors, were place of residence, region, community place of delivery, community wealth, community media exposure, and community toilet facility. The individual-level predictors were further categorized as socio-demographic and economic factors like the educational level of the mother, sex of the household head, age of household head, number of household members, number of children under the age of five, marital status of the mother, source of water, time to get water, type of toilet facility, household electricity, types of cooking fuel, main floor material, main wall material, and main roof material and maternal and child factors (such as maternal age at first birth, sex of the child), utilization of contraception, order of birth, mode of delivery, duration of breastfeeding and multiple births (Fig 1).

### Spatial analysis

The data for spatial analysis was cleaned and merged using STATA version 16 and Microsoft Excel. ArcGIS version 10.8 and saTScan version 9.7 were used for the spatial analysis.

### Spatial autocorrelation

Spatial autocorrelation (Global Moran's I) analysis was conducted to examine whether under-five mortality was dispersed (Moran's I value closer to -1), clustered (Moran's I value closer to 1), or randomly distributed (Moran's I value of 0) in Ethiopia [22].

### Spatial interpolation

The under-five mortality was known in enumerated areas, while in areas that were not selected, the under-five mortality rates were predicted. Spatial interpolation was applied using the geostatistical ordinary Kriging spatial interpolation technique to predict under-five mortality from existing sample data points to un-sampled areas [23].

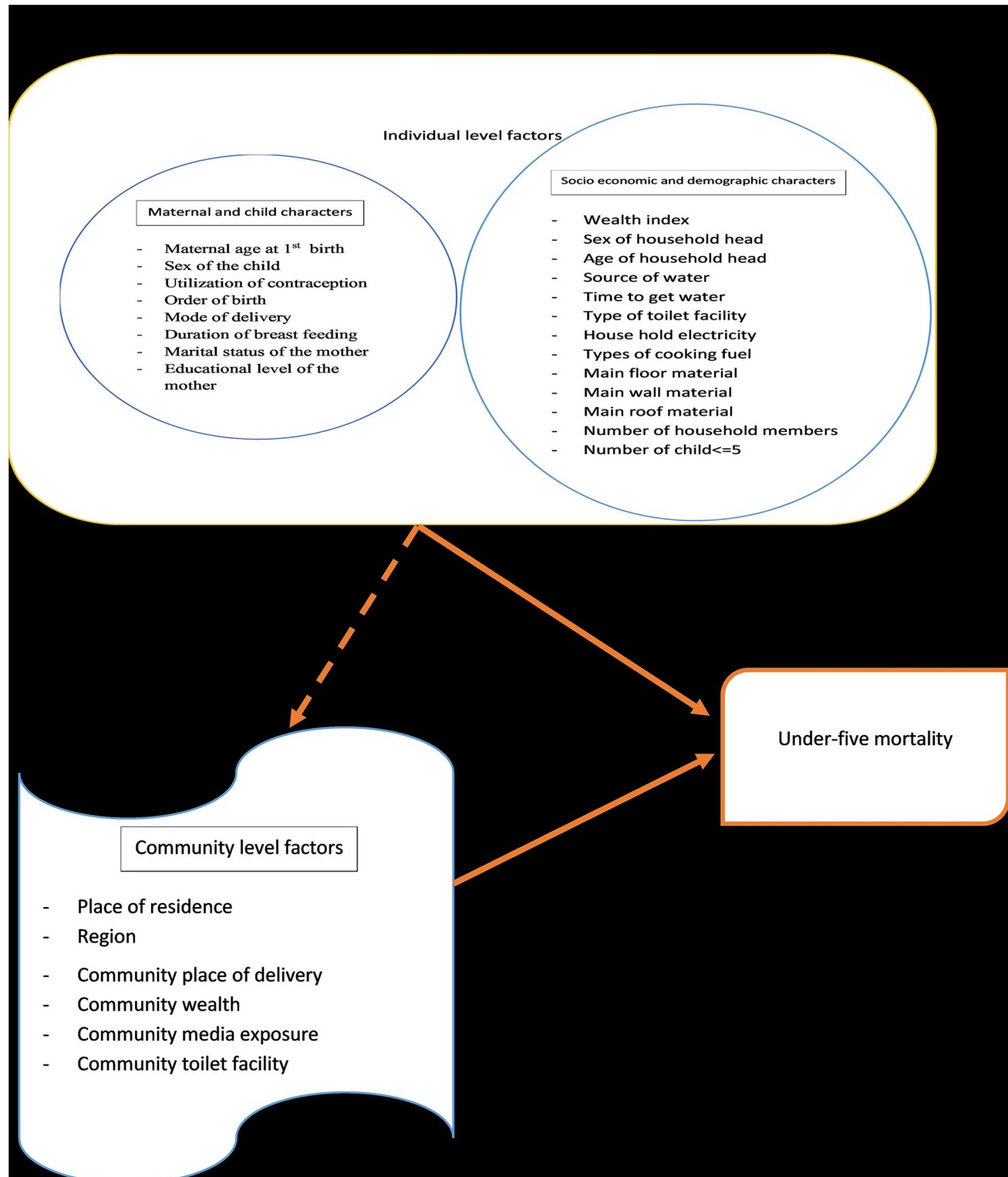

**Fig 1. Conceptual framework of under-five mortality in Ethiopia 2019.**

## Spatial scan statistics

The scan analysis was performed using SaTscan, based on the Bernoulli test for cases (child is not alive) and controls (child is alive). The upper limit used was the default maximum spatial cluster size of less than 50% of the population, allowing both small and large clusters to be detected, while clusters that contained more than the maximum limit with the circular shape of the window were avoided [24]. Most likely clusters were identified using p-values and likelihood ratio tests, which is the ratio of the likelihood of the alternative hypothesis (higher activity level inside the window) over the likelihood of the null hypothesis (same activity level inside and outside).

## Data management and analysis

We used STATA version 16 and R statistical software version 4.0.5 to analyze the data. A total of 31 variables were retained for the analysis. Residents who were not De jure were excluded which affected under-five mortality, as they could not respond to most of the socio-demographic and economic characteristics even though they could answer the maternal and child characteristics. This exclusion changed our sample size from 5,753 to 5,695.

The outcome variable was re-coded to (child alive: Yes = 1 and No = 0). Four community variables were generated by taking the individual variable, calculating their proportion, and dichotomizing them based on their mean or median according to their distribution.

In the end, we had 29 predictor variables, of which 6 were community-level predictor variables and 23 were individual-level predictor variables.

Based on EMDHS, respondents in the same cluster showed similar outcomes or functions at the same level and the data has a hierarchical structure. This made binary logistic regression not the most appropriate as it violates the assumption of independence of the residuals. Instead, a model that considers clustering effect should be used [7, 10]. Multilevel logistic modeling separates the within-cluster effects from the between-cluster effects [25]. Therefore, to assess the predictors associated with U5M, a non-weighted multilevel logistic regression model was used.

Bivariable multilevel logistic regression was used to screen each predictor variable for a p-value less than 0.2. Significant variables were included in multivariable multilevel logistic models. Twenty predictor variables (3 of which were community-level predictor variables) were included in the multivariable analysis. In the multivariable analysis, a p-value less than 0.05 was considered a factor associated with U5M.

The first model fitted was the null model (intercept model), which contained the outcome variable only (under-five mortality status) with the cluster number. The intra-cluster correlation (ICC) was used to assess whether there was a random effect. An ICC of 0.130 which meant there was a minimum of 13% under-five mortality was explained by between-cluster differences. We found that 87% of under-five mortality was explained by within-cluster differences, which was not negligible. The second model was fitted using the outcome, the cluster and the individual-level predictor variables only. The probability of U5M was predicted as a function of individual-level predictors.

For the third model the outcome variable, the cluster number, and the community-level predictor variables were accounted for. Then the final model was fitted by taking both the individual-level and the community-level predictor variables into account.

The models were compared by using a log-likelihood statistic, where the best model was selected based on smallest deviance.

### Ethical consideration

Permission for data access was obtained from a major demographic and health survey through an online request from http://www.dhsprogram.com. The data used for this study were publicly available with no personal identifiers.

## Results

### Socio-demographic background and maternal and child health factor

A total of 5,695 live births were included in this study, of which 2,942 (51.66%) were male, 5,528 (97.07%) were single children, 2,937 (51.57%) were delivered at home and 6.04% of the children were born via cesarean section (Table 1).

Out of the total children's mothers, most were married (5,346, 93.87%) and 3,135 (55.05%) had no education, only 291 (5.11%) attained higher education, and 51.62% were of poor economic status. 64.27% had access to improved water; 30.99% travel more than 30 minutes, and 50.85% travel less than 30 min to get water because they do not have water on-premises. The majority of household heads were male, 4,561 (80.09%), and the most common age ranges of the household head was 35–50 years (Table 1).

At the community level, the majority were rural residents 4,389 (77.07%), and 3,457 (60.70%) in the community practiced home delivery. The number of communities that used unimproved toilets was 3,175 (55.75%). And 3,391 residents of the population (59.54%) were not exposed to the media (Table 2).

### Prevalence of under-five mortality

The under-five mortality was 5.9% (95% CI 5.4%, 6.6%) and it varied across regions ranging from 2.13% in Addis Ababa to 9.82% in the Somali region. Out of the total, there were 918 (15.79%) children who died having been breastfed for less than 6 months (Table 3).

### Spatial autocorrelation

The global Moran's index was 0.09, z-score 3.5, and p-value <0.001, suggesting that the spatial distribution under-five mortality was not at random in the 2019 EMDHS (Fig 2).

### Hotspot analysis

A high proportion of U5M was observed around the regions of Somali and Dire Dawa (Fig 3) [26].

### Spatial interpolation

The spatial interpolation or prediction analysis result showed that the under-five mortality ranged from 0% to 50%. The areas predicted to have the highest risk were located in Somali, Dire Dawa, Benishangul Gumuz, Gambella, and SNNPR (Fig 4) [26].

### Spatial scan statistics analysis

As observed in Fig 5, significant clusters were identified in the Somali region. Located within 7.384788 N, 45.908936 E with 612.75 km radius. Children living in the primary cluster were 62% more likely to die before they celebrate their fifth birthday when compared to those who lived outside the window (relative risk (1.62), likelihood ratio (7.36), and p-value (0.03) (Fig 5) [26].

**Table 1.  Sociodemographic, maternal, and child health factors.**

| Sociodemographic, Maternal and child health factors | | Frequency | Percentage |
|---|---|---|---|
| Age of the mother at first birth | Less than 18 | 3,024 | 53.10% |
| | 19 to 34 | 2,650 | 46.53% |
| | 35 and more | 21 | 0.37% |
| Sex of child | Male | 2,942 | 51.66% |
| | Female | 2,753 | 48.34% |
| Current Contraceptive utilization | Not used | 3,875 | 68.04% |
| | Used | 1,820 | 31.96% |
| Multiple births | Singleton | 5,528 | 97.07% |
| | Multiple | 167 | 2.93% |
| Birth order | 1st order | 1,228 | 21.56% |
| | 2nd and 3rd order | 1,883 | 33.06% |
| | 4th and 5th order | 1,298 | 22.79% |
| | 6th order + | 1,286 | 22.58% |
| Place where the mother delivered | Home delivery | 2,937 | 51.57% |
| | Facility delivery | 2,758 | 48.43% |
| Delivery by cesarean section | No | 5,351 | 93.96% |
| | Yes | 344 | 6.04% |
| Months of Breastfeeding | Less than 6 months | 918 | 16.12% |
| | 6 months and more | 4,777 | 83.88% |
| Sex of household head | Male | 4,561 | 80.09% |
| | Female | 1,134 | 19.91% |
| Age of household head (years?) | Less than 35 | 2,479 | 43.53% |
| | 35 to 50 | 2,596 | 45.58% |
| | 50 and more | 620 | 10.89% |
| Marital Status | Married | 5,346 | 93.87% |
| | Unmarried | 349 | 6.13% |
| Highest educational level | No education | 3,135 | 55.05% |
| | Primary | 1,802 | 31.64% |
| | Secondary | 467 | 8.20% |
| | Higher | 291 | 5.11% |
| Wealth index | Poor | 2,940 | 51.62% |
| | Middle | 797 | 13.99% |
| | Rich | 1,958 | 34.38% |
| Water source | Improved | 3,660 | 64.27% |
| | Unimproved | 2,035 | 35.73% |
| Time to fetch water | More than 30 min | 1,765 | 30.99% |
| | Less than 30 min | 2,896 | 50.85% |
| | On premises | 1,034 | 18.16% |
| Type of latrine used | Improved | 1,164 | 20.44% |
| | Not improved | 4,531 | 79.56% |
| Household has electricity | No | 4,073 | 71.52% |
| | Yes | 1,622 | 28.48% |
| Cooking fuel in use | Clean fuel | 432 | 7.59% |
| | Solid fuel | 5,263 | 92.41% |
| Type of floor of the house | Earth/sand/dung | 4,384 | 76.98% |
| | Wood | 49 | 0.86% |
| | Cement | 799 | 14.03% |
| | Carpet | 463 | 8.13% |

(*Continued*)

**Table 1.** (Continued)

| Sociodemographic, Maternal and child health factors | | Frequency | Percentage |
|---|---|---|---|
| Type of the house wall | Cane/Palm/Trunk/Reed | 1,261 | 22.14% |
| | Bamboo/wood | 3,531 | 62.00% |
| | Cement/stone with lime/bricks/covered adobe | 903 | 15.86% |
| Type of the house roof | Thatch | 1,488 | 26.13% |
| | Palm/bam | 995 | 17.47% |
| | Cement | 198 | 3.48% |
| | Corrugated iron | 3,014 | 52.92% |

**Table 2. Community-level variable.**

| Community level variable | | frequency | Percentage |
|---|---|---|---|
| Place of residence | Urban | 1,306 | 22.93% |
| | Rural | 4,389 | 77.07% |
| Community place of delivery | Home delivery | 3,457 | 60.70% |
| | Facility delivery | 2,238 | 39.30% |
| Community toilet utilization | Improved | 2,520 | 44.25% |
| | Not improved | 3,175 | 55.75% |
| Community wealth status | Poor | 2,776 | 48.74% |
| | Rich | 2,919 | 51.26% |
| Community media exposure | Not exposed to media | 3,391 | 59.54% |
| | Exposed to media | 2,304 | 40.46% |

**Table 3. Under-five mortality among predictors.**

| Characteristics | | Is the child alive? | | total |
|---|---|---|---|---|
| | | Yes 5357(94.06%) | No338(5.94%) | |
| Is child twin? | Yes | 123(2.16%) | 44(0.77%) | 167(2.93%) |
| | No | 5234(91.9%) | 294(5.16%) | 5,528(97.06%) |
| Region | Tigray | 436(7.65%) | 14(0.24%) | 450(7.90%) |
| | Afar | 616(10.81%) | 35(0.612%) | 651(11.43%) |
| | Amhara | 478(8.39%) | 23(0.40%) | 501(8.79%) |
| | Oromia | 674(11.83%) | 41(0.72%) | 715(12.55%) |
| | Somali | 580(10.18%) | 57(1.00%) | 637(11.18%) |
| | Benishangul | 477(8.37%) | 45(0.79%) | 522(9.17%) |
| | SNNPR | 631(11.08%) | 26(0.46%) | 657(11.54%) |
| | Gambella | 409(7.18%) | 35(0.61%) | 444(7.79%) |
| | Harari | 408(7.16%) | 29(0.51%) | 437(7.67%) |
| | Addis Ababa | 281(4.93%) | 6(0.11%) | 287(5.04%) |
| | Dire Dawa | 367(6.44%) | 27(0.47%) | 394(6.92%) |
| Breastfeeding | Less than 6 months | 773(13.57%) | 145(2.55%) | 918(16.12%) |
| | 6 months and more | 4,584(80.49%) | 193(3.39%) | 4,777(83.88%) |
| Birth order | 1st order | 1,142(20.05%) | 86(1.52%) | 1,228(21.56%) |
| | 2-3rd order | 1,793(31.48%) | 90(1.56%) | 1,883(33.06%) |
| | 4-5th order | 1,231(21.62%) | 67(1.18%) | 1,298(22.79%) |
| | 6th + order | 1,191(20.91%) | 95(1.67%) | 1,286(22.58%) |
| Place of delivery | Home delivery | 2,733(47.99%) | 204(3.58%) | 2,937(51.57%) |
| | Facility delivery | 2,624(46.06%) | 134(2.35%) | 2,758(48.43%) |

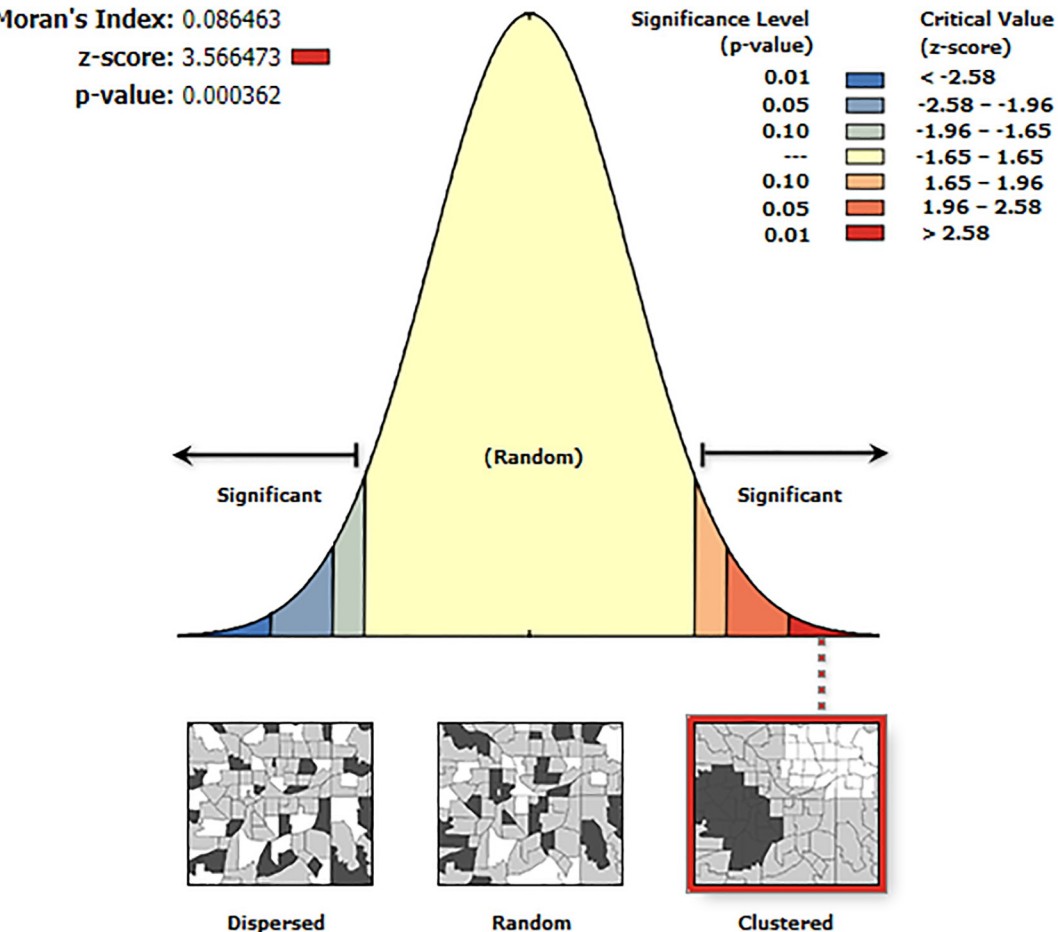

**Fig 2. Spatial autocorrelation of under-five mortality in Ethiopia 2019.** Source: https://open.africa/dataset/africa-shapefiles).

## Factors associated with under-five mortality

After filtering, all variables in bivariable multilevel logistic regression, 4 models were fitted (null model, individual-level model, community-level model, and both individual- and community-level model). The ICC was 0.13 with 95% CI: 0.08 to 0.2, which is not negligible. The fourth model was the best-fitting model with the lowest deviance. When the number of household members increased by one, the odds of under-five mortality decreased by 8.5% (AOR = 0.92, 95% CI: 0.84, 0.99). The odds of under-five death decreased by 82.8% (AOR = 0.17, 95% CI: 0.14, 0.21) as the number of under-five children in the family increased by one. Children who were part of multiple births had 14 times higher odds of under-five mortality (AOR = 14.4, 95% CI: 8.5, 24.3) than singletons. Children who were breastfed for less than 6 months had 5 times higher odds of experiencing under-five mortality (AOR = 5.04, 95% CI: 3.81, 6.67) than those who were breastfed. People whose main roof was palm/bamboo had 43% (AOR = 0.57, 95% CI: 0.34, 0.96) decreased odds of having under-five deaths in the family than those with corrugated iron. Under-five children who were the sixth born or more had 2.5 (AOR = 2.46, 95% CI: 1.49, 4.06) times higher odds of under-five mortality compared to the second and third born. Place of delivery was found to have a significant association with

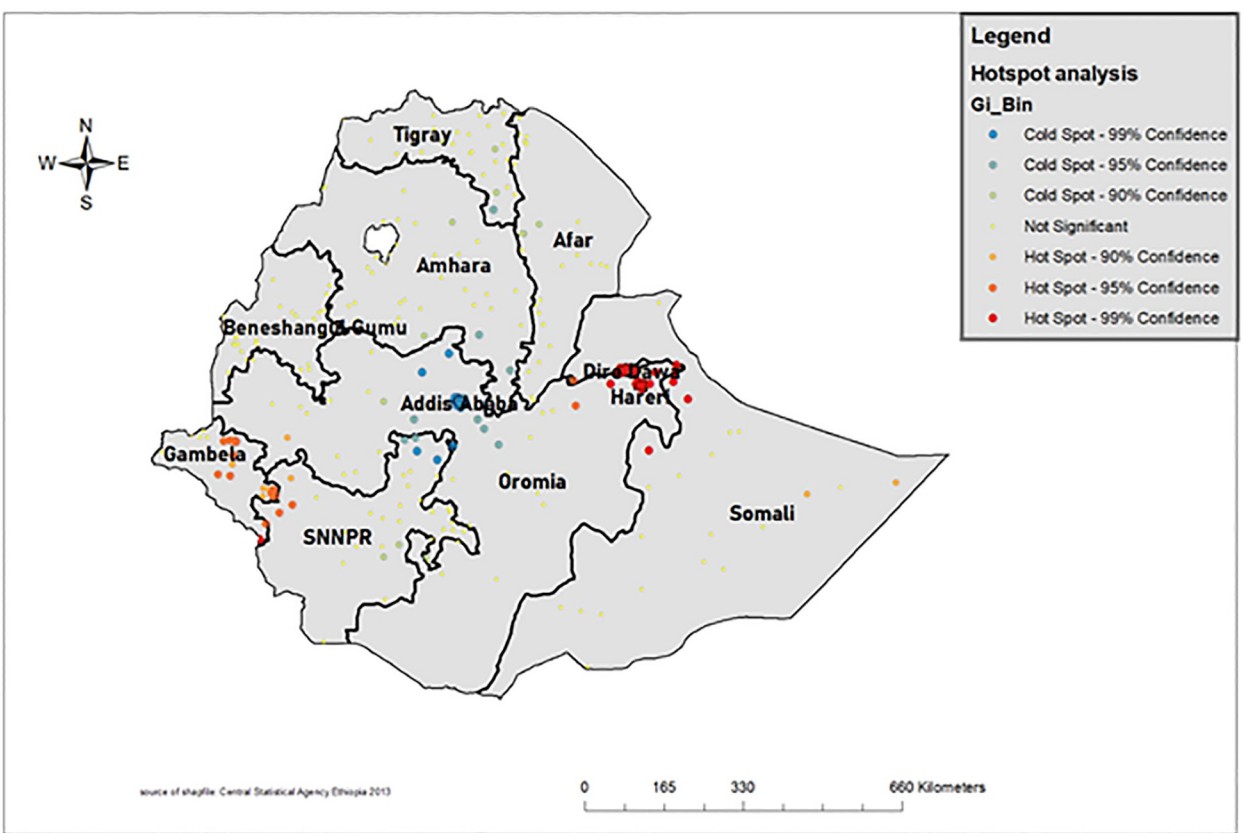

**Fig 3. Hotspot analysis of under-five mortality in Ethiopia 2019.** (Source: https://open.africa/dataset/africa-shapefiles).

U5M; those who were delivered in institutions had 43% less chance of experiencing U5M (AOR = 0.57, 95% CI: 0.41, 0.81) compared to those delivered at home. Being a resident of the Somali and Afar region increased the odds of under-five death by 3.5 (AOR = 3.46, 95% CI: 1.58, 7.55) and 2.5 (AOR = 2.54, 95% CI: 1.10, 5.85) times, respectively, compared to the Tigray region (Table 4).

The residual plot shows that the model performed adequately, with the Kolmogorov smirnov (KS) test being insignificant (p-value = 0.29), and also the deviation was not significant (Fig 6).

## Discussion

The study mainly focused on the spatial distribution of under-five mortality and on factors that influence in Ethiopia. The findings of this study revealed that the prevalence of under-five mortality was 5.9% (95% CI: 5.4, 6.6). The multivariable multilevel logistic regression model result showed that family size, number of under-five children, birth order, breastfeeding, multiple birth, region, and institutional delivery were associated with under-five mortality in Ethiopia. The spatial analysis result revealed that Somali, Dire Dawa and Eastern Oromia were high-risk areas for under five mortality in Ethiopia.

Increased number of family size resulted in decreased odds of under-five death. This finding is consistent with previous studies conducted in Ethiopia [16], Eastern Nigeria [27], and Ghana [28]. This may be due to the mother of the children being more experienced in taking

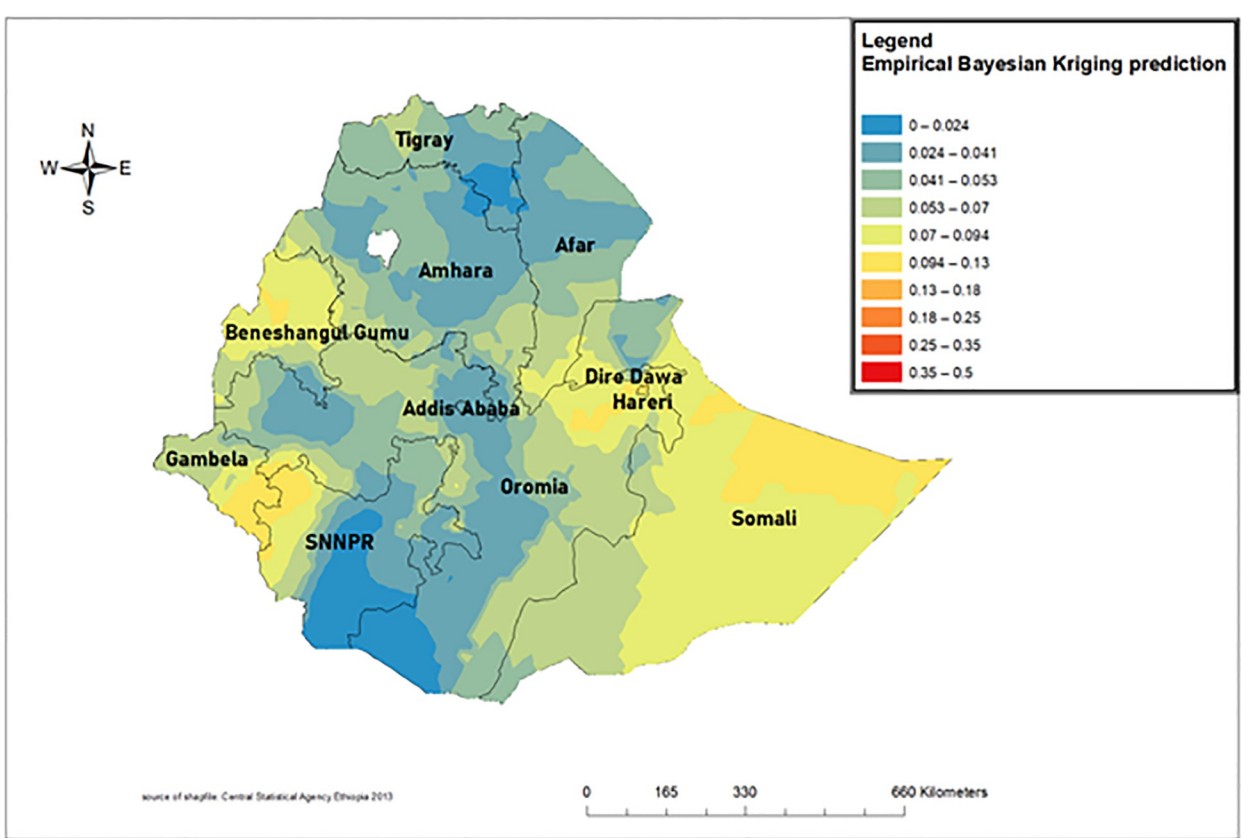

**Fig 4. Empirical bayesian kriging prediction map of under-five mortality in Ethiopia 2019.** (Source: https://open.africa/dataset/africa-shapefiles).

good and adequate care of the children as the family size increases, in addition to there being more family members that can care for under-five children. However, this is in contrast to another study conducted in Ethiopia, which demonstrated that the odds of under-five mortality increased as the number of under-five children increased [18]. This may indicate that, as the number of under-five children increases, it may become difficult to fulfill their basic needs, and family exhausted to look after the children.

Birth order had significant effect on under-five mortality. This study indicates that, as order of birth increased, the likelihood of child death also increased. This finding is consistent with a study conducted in rural parts of Ethiopia [9]. However, it does not support another study conducted in Ethiopia which found that being firstborn increased the odds of under-five mortality twofold when compared to those with five and above birth orders [18]. This may be because firstborns are associated with younger and less experienced mothers, which may be related to increased mortality. In addition, firstborns are more likely to be hospitalized as a result of congenital malformations and perinatal conditions in their early childhood [29]. This changes for firstborn children as they get older, while younger brothers and sisters are more likely to be hospitalized for injuries and avoidable conditions, indicating less parental care [30].

Being breastfed for less than six months increased the odds of under-five mortality compared with those who were breastfed for six and more months. This is consistent with a finding in Ethiopia [18]. Breastfeeding is known as a preventive method for reducing child mortality and as well as a prevention for delayed growth [31, 32].

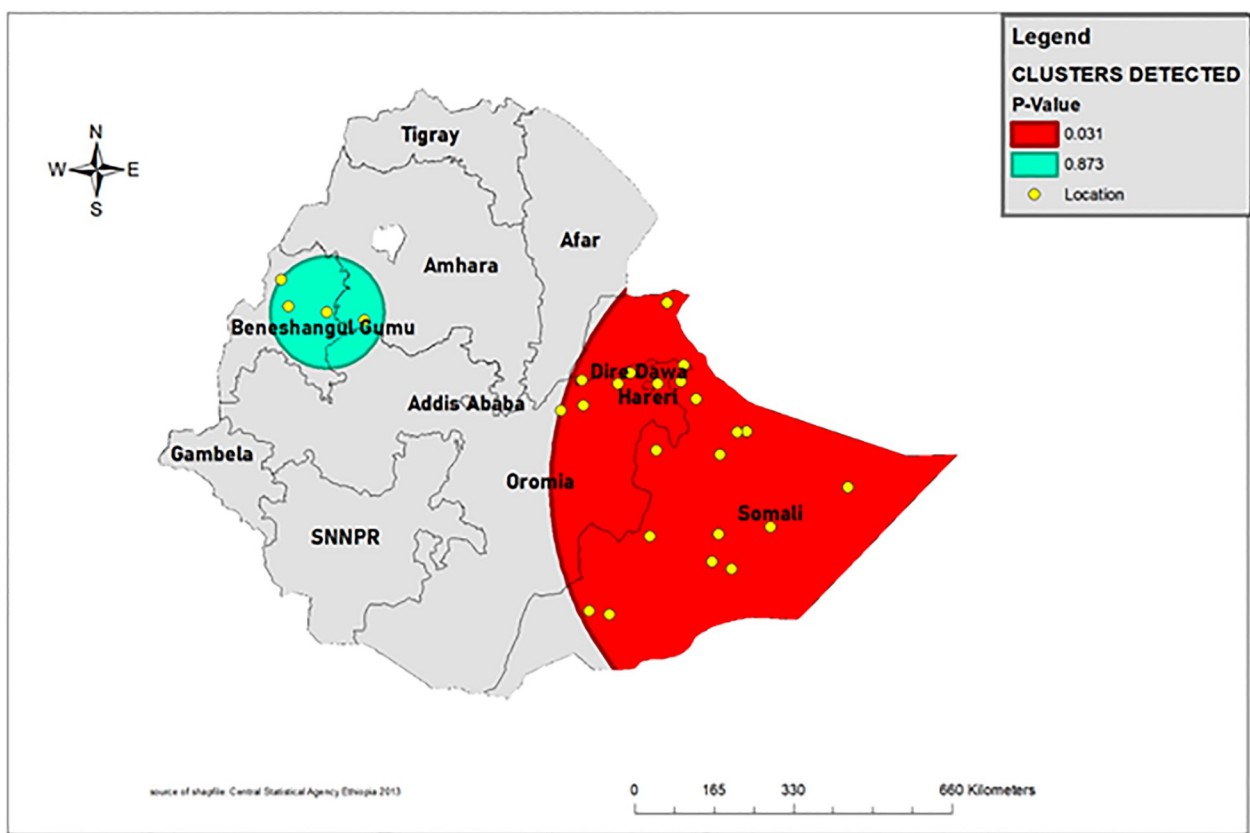

**Fig 5. Spatial scan statistics of under-five mortality in Ethiopia 2019.** (Source: https://open.africa/dataset/africa-shapefiles).

Being multiple birth children increased the odds of under-five mortality when compared to singleton children. This finding is in line with several studies conducted in Ethiopia at different times, and study conducted in rural parts of Ethiopia [6, 7, 9, 16, 18]. This may be due to multiple births imposing short- and long-term medical complications as well as adding increased burden on the mother and the family [33].

Facility or institutional delivery reduced the odds of under-five mortality by 43% as compared to home delivery. This finding is consistent with studies conducted in Ethiopia [9, 10] and the Democratic Republic of Congo [13]. This is because home delivery increases the death of under-five children due to a higher risk of intrapartum and postpartum complications. These complications include exposure to common vaccine-preventable diseases if there is absence of vaccination, and a lower likelihood of attending prenatal and postpartum visits [10, 34].

The odds of under-five mortality were higher for residents of Afar and Somali regions compared to the Tigray region. This coincides among other things, with Somali and Afar regions having the first and second lowest coverage of children with all basic as well as second and first prevalence respectively [35, 36]. Additionally, the Somali region was one of the regions that were in the primary cluster from our spatial scan statistics analysis where under-five mortality is more likely to occur. This is not supported by studies conducted in Ethiopia based on the 2011 and 2016 EDHS [9, 10], where other regions were found to have a significant association with under-five mortality. This discrepancy may be a result of the different number of enumerated areas (clusters) used in the different studies, which may have led to different samples, or different statistical models used for analysis.

**Table 4. Multivariable multilevel logistic regression analysis of both individual and community-level factors associated with under-five mortality in Ethiopia, EMDHS 2019.**

| Variable | Null model | Model2 AOR (95%CI) | Model3 AOR(95%CI) | Model4 AOR(95%CI) |
|---|---|---|---|---|
| Education | | | | |
| No education | | 1 | | 1 |
| Primary | | 1.14 (0.83,1.56) | | 1.14 (0.83,1.58) |
| Secondary | | 0.59 (0.31, 1.13) | | 0.58(0.30,1.12) |
| Higher | | 0.65(0.28, 1.55) | | 0.60(0.25,1.44) |
| Household members | | 0.922(0.85, 1.01) | | 0.92(0.83,0.99) * |
| Number of under-five children | | 0.17 (0.14, 0.22) | | 0.17(0.14, 0.21) *** |
| Sex | | | | |
| Male | | 1 | | 1 |
| female | | 0.83(0.64, 1.07) | | 0.84(0.64,1.09) |
| Mode of delivery by C/S | | | | |
| No | | 1 | | 1 |
| Yes | | 1.36 (0.79, 2.35) | | 1.37(0.79,2.39) |
| Preceding birth order | | | | |
| 1st | | 1 | | 1 |
| 2-3rd | | 1.32(0.92, 1.91) | | 1.34(0.93,1.94) |
| 4-5th | | 1.52(0.97, 2.39) | | 1.51(0.96,2.38) |
| 6+ | | 2.44(1.48, 4.02) | | 2.46(1.49,4.06) *** |
| Time to water | | | | |
| >30min | | 1 | | 1 |
| <30min | | 0.77(0.56, 1.04) | | 0.89(0.64,1.23) |
| On-premises | | 0.66(0.38, 1.14) | | 0.76(0.43,1.34) |
| Cooking fuel | | | | |
| Clean fuel | | 1 | | 1 |
| Solid fuel | | 1.27(0.62, 2.58) | | 0.93(0.45,1.94) |
| Age at 1st birth | | | | |
| 19–34 | | 1 | | 1 |
| < = 18 | | 0.91(0.70, 1.20) | | 0.90(0.68,1.18) |
| > = 35 | | 2.58(0.55, 12.01) | | 2.72(0.55,13.59) |
| Breastfeeding | | | | |
| < 6 months | | 1 | | 1 |
| > = 6 months | | 5.09(3.85, 6.72) | | 5.05(3.82,6.67) *** |
| Place of delivery | | | | |
| Home delivery | | 1 | | 1 |
| Facility delivery | | 0.54 (0.40,0.75) | | 0.57(0.41,0.81) ** |
| Main floor | | | | |
| earth/sand/dung | | 1 | | 1 |
| Wood | | 1.71(0.53, 5.52) | | 2.14(0.66,6.95) |
| Cement | | 0.89(0.50, 1.59) | | 0.86(0.48,1.58) |
| Carpet | | 1.56(0.85, 2.86) | | 1.28(0.67,2.46) |
| Main wall | | | | |
| bamboo/wood | | 1 | | 1 |
| cane/palm/trunks/re ed | | 1.42(0.97, 2.07) | | 1.06(0.70,1.58) |
| cement/stone | | 1.53(0.89,2.61) | | 1.34(0.76,2.34) |
| Main roof | | | | |
| Corrugated iron | | 1 | | |

(*Continued*)

**Table 4.** (Continued)

| Variable | Null model | Model2 AOR (95%CI) | Model3 AOR(95%CI) | Model4 AOR(95%CI) |
|---|---|---|---|---|
| thatch | | 1.25(0.87,1.79) | | 1.14(0.77,1.67) |
| palm/bamboo | | 0.71(0.44,1.15) | | 0.57(0.34,0.96)* |
| calamine/cement | | 1.14(0.54,2.44) | | 1.04(0.47,2.29) |
| Contraceptive utilization | | | | |
| No | | 1 | | 1 |
| Yes | | 0.72(0.53,0.98) | | 0.82(0.60,1.13) |
| Marital status | | | | |
| Currently Married | | 1 | | 1 |
| Currently Unmarried | | 1.19(0.74,1.92) | | 1.22(0.75,1.97) |
| Multiple Twin child | | | | |
| No | | 1 | | 1 |
| Yes | | 15.63(9.33,26.20) | | 14.39(8.51,24.33)*** |
| Region | | | | |
| Tigray | | | 1 | 1 |
| Afar | | | 1.60(0.76,3.41) | 2.55(1.10,5.85)* |
| Amhara | | | 1.47(0.68,3.19) | 0.97(0.44,2.15) |
| Oromia | | | 1.73(0.84,3.55) | 1.52(0.73,3.15) |
| Somali | | | 2.86(1.39,5.89) | 3.46(1.59,7.55)** |
| Benishangul gumuz | | | 2.74(1.32,5.68) | 1.95(0.91,4.20) |
| SNNPR | | | 1.17(.55,2.47) | 0.94(0.44,2.03) |
| Gambella | | | 2.71(1.29,5.69) | 1.94(0.90,4.21) |
| Harari | | | 2.37(1.12,5.04) | 1.95(0.87,4.36) |
| Addis Ababa | | | 0.71(0.25,2.03) | 0.64(0.21,2.02) |
| Dire Dawa | | | 2.18(1.01,4.70) | 2.02(0.91,4.50) |
| Community delivery place | | | | |
| Home delivery | | | 1 | 1 |
| Facility delivery | | | 0.75(0.52, 1.09) | 0.94(0.62,1.43) |
| Community media | | | | |
| Not exposed | | | 1 | 1 |
| exposed | | | 1.09(0.75,1.58) | 0.94(0.64,1.40) |
| Intercept | 0.49(0.30,0.81) | 0.22(0.08,0.63) | 0.35(0.19,0.64) | 0.16(0.04,0.57) |
| ICC | 0.130 | 0.036 | 0.091 | 0.025 |
| Log-likelihood | -1263.56 | -918.15 | -1247.40 | -903.72 |

*** P-value<0.001,

** p-value<0.01,

* p-value<0.05

## Conclusion

Under-five mortality in Ethiopia was 5.9% of livebirth. Being breastfed for more than 6 months, twin birth, institutional delivery and high-risk areas of U5M (Somali and dire Dawa) were modifiable risk factors. Therefore, maternal and community education on the advantage of breastfeeding and institutional delivery is highly recommended. Women who deliver multiple births should be given special attention. Further, effective strategies should be designed for interventions on hot spot areas of U5M, namely Somali and Dire Dawa.

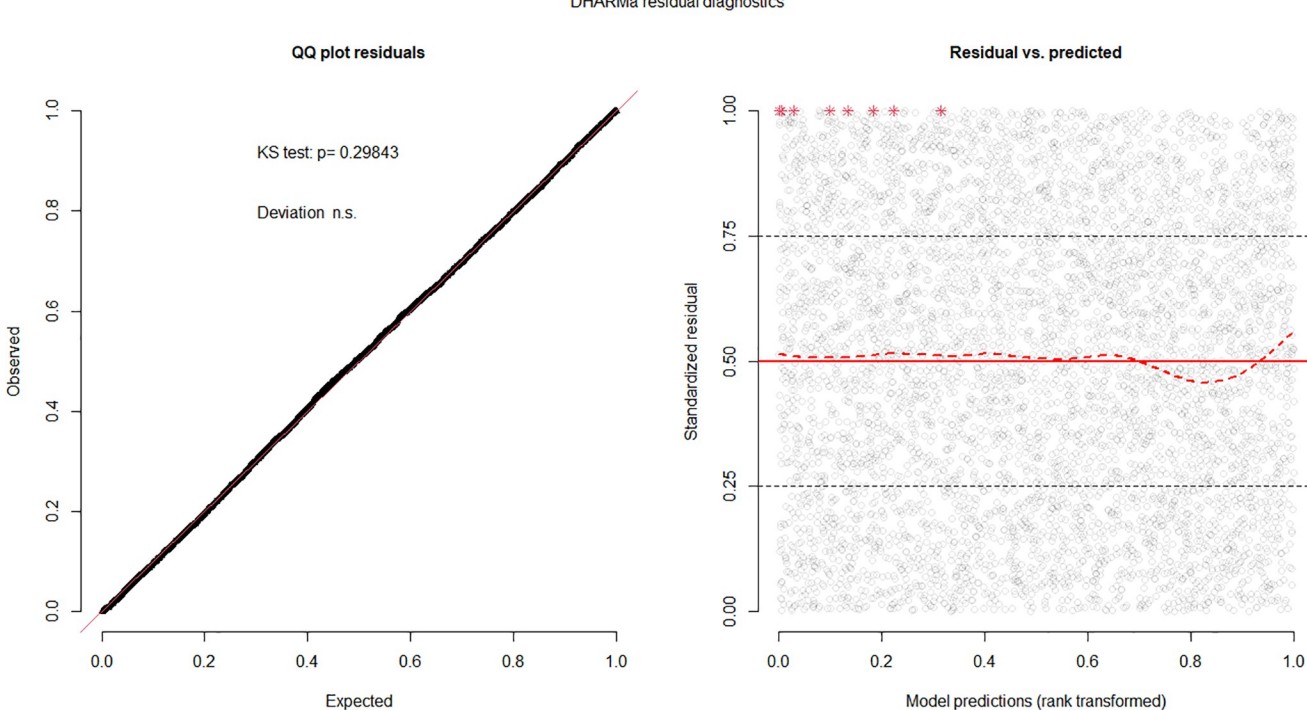

**Fig 6. residual diagnostic plot for the final model of under-five mortality in Ethiopia 2019.**

## Strengths and limitations of the study

We used EMDHS 2019 data which is more inclusive and generalizable as Enumeration Areas (EAs) are taken from most parts of the country. The larger sample size increased the power of the study and helped to make geographical comparisons. As this was cross-sectional, factors associated are not necessarily of causal effects, since results do not show a temporal relationship. Additionally, potentially important variables such as antenatal care visits, postnatal care, and preceding birth interval were excluded from the study due to their high rate of missing values, hence the study may missing important predictive factors associated with U5M.

## Acknowledgments

We acknowledge the DHS program for providing us with the dataset. Elizabeth Korevaar reviewed the English language of the manuscript.

## Author Contributions

**Conceptualization:** Zemenu Tadesse Tessema, Tsion Mulat Tebeje, Lewi Goytom Gebrehewet.

**Data curation:** Zemenu Tadesse Tessema, Tsion Mulat Tebeje, Lewi Goytom Gebrehewet.

**Formal analysis:** Zemenu Tadesse Tessema, Tsion Mulat Tebeje, Lewi Goytom Gebrehewet.

**Funding acquisition:** Zemenu Tadesse Tessema, Tsion Mulat Tebeje, Lewi Goytom Gebrehewet.

**Investigation:** Zemenu Tadesse Tessema, Tsion Mulat Tebeje, Lewi Goytom Gebrehewet.

**Methodology:** Zemenu Tadesse Tessema, Tsion Mulat Tebeje, Lewi Goytom Gebrehewet.

**Project administration:** Zemenu Tadesse Tessema, Tsion Mulat Tebeje, Lewi Goytom Gebrehewet.

**Resources:** Zemenu Tadesse Tessema, Tsion Mulat Tebeje, Lewi Goytom Gebrehewet.

**Software:** Zemenu Tadesse Tessema, Tsion Mulat Tebeje, Lewi Goytom Gebrehewet.

**Supervision:** Zemenu Tadesse Tessema, Tsion Mulat Tebeje, Lewi Goytom Gebrehewet.

**Validation:** Zemenu Tadesse Tessema, Tsion Mulat Tebeje, Lewi Goytom Gebrehewet.

**Visualization:** Zemenu Tadesse Tessema, Tsion Mulat Tebeje, Lewi Goytom Gebrehewet.

**Writing – original draft:** Zemenu Tadesse Tessema, Tsion Mulat Tebeje, Lewi Goytom Gebrehewet.

**Writing – review & editing:** Zemenu Tadesse Tessema, Tsion Mulat Tebeje, Lewi Goytom Gebrehewet.

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
