## [Decision Letter · Decision Letter 0]

7 Dec 2021

PONE-D-21-20589Geographic variation and factors associated with under-five mortality in Ethiopia. A spatial and multilevel analysis of Ethiopian Mini Demographic and Health Surveys 2019.PLOS ONE

Dear Dr. Tessema,

Thank you for submitting your manuscript to PLOS ONE. After careful consideration, we feel that it has merit but does not fully meet PLOS ONE’s publication criteria as it currently stands. Therefore, we invite you to submit a revised version of the manuscript that addresses the points raised during the review process.

Your manuscript has been reviewed by 8 reviewers who have made suggestions regarding the study rationale, statistical analysis, and discussion. In addition, the language in the manuscript requires editing.

We look forward to receiving your revised manuscript.

Kind regards,

Nancy Beam, PhD

Staff Editor

PLOS ONE

“We didn’t receive external funds for this research.”

5. Your ethics statement should only appear in the Methods section of your manuscript. If your ethics statement is written in any section besides the Methods, please delete it from any other section. "

6. We note that Figures 1-4 in your submission contain [map/satellite] images which may be copyrighted. All PLOS content is published under the Creative Commons Attribution License (CC BY 4.0), which means that the manuscript, images, and Supporting Information files will be freely available online, and any third party is permitted to access, download, copy, distribute, and use these materials in any way, even commercially, with proper attribution. For these reasons, we cannot publish previously copyrighted maps or satellite images created using proprietary data, such as Google software (Google Maps, Street View, and Earth). For more information, see our copyright guidelines: http://journals.plos.org/plosone/s/licenses-and-copyright.

a. You may seek permission from the original copyright holder of Figures 1-4 to publish the content specifically under the CC BY 4.0 license. 

Reviewers' comments:

Reviewer's Responses to Questions

**Comments to the Author**

1. Is the manuscript technically sound, and do the data support the conclusions?

Reviewer #1: Yes

Reviewer #2: Partly

Reviewer #3: Yes

Reviewer #4: Yes

Reviewer #5: Yes

Reviewer #6: Yes

Reviewer #7: Yes

Reviewer #8: Partly

2. Has the statistical analysis been performed appropriately and rigorously? 

Reviewer #1: Yes

Reviewer #2: I Don't Know

Reviewer #3: Yes

Reviewer #4: Yes

Reviewer #5: Yes

Reviewer #6: Yes

Reviewer #7: Yes

Reviewer #8: No

3. Have the authors made all data underlying the findings in their manuscript fully available?

Reviewer #1: Yes

Reviewer #2: Yes

Reviewer #3: Yes

Reviewer #4: Yes

Reviewer #5: Yes

Reviewer #6: Yes

Reviewer #7: Yes

Reviewer #8: Yes

4. Is the manuscript presented in an intelligible fashion and written in standard English?

Reviewer #1: Yes

Reviewer #2: No

Reviewer #3: Yes

Reviewer #4: No

Reviewer #5: No

Reviewer #6: No

Reviewer #7: Yes

Reviewer #8: Yes

5. Review Comments to the Author

Reviewer #1: Overall authors have explained regional variation in under five mortality and the key predictors of U5M in Ethiopia. I have few comments to improve this paper further.

1. Authors have used random intercept model for binary multilevel logistic regression. Authors can also do some more testing of their model by including random slopes for their significant predictors in this model to check if the effect of their key predictors remains the same for each cluster or is it different. If they don't found any major difference by including the random slopes, then they can provide the current results as it is.

2. Authors have used different household condition indicators such as toilet clean fuel main roof etc than wealth index which captures the economic condition of the household better in the analysis. Is there any specific reason to adapt this strategy?

3. Authors should also provide the residuals plots for their final multilevel model in the appendix.

4. Authors have shown that there is a significant large cluster of population where U5MR is significantly higher than other areas in Figure 4. Authors can include a table with basic socio demographic profile of this high cluster region. This can be helpful to understand which population mostly suffers with high U5M in that region so that specific interventions and programmes can be targeted for those subgroups of population.

5. Authors should include a short paragraph in the discussion on the implications of their findings in context of the SDG goals.

6. Please see the flow of your texts in the discussion once more and edit the sentences wherever required for more clarity.

Reviewer #2: This publication is of high importance because it attempts to present sophisticated analyses of a legitimate population-based survey data to highlight key risk factors for under five mortality in Ethiopia. However, there are two problems with the manuscript: the first is that it is poorly written in English, which makes understanding of all processed hard; secondly, there are some explanations about the statistics used that (partly due to the poor English) are difficult to understand. Double check both the English and the statistical methods used (binary, bivariable?, multi-level logistic regression?).

Reviewer #3: This is an engaging, insightful paper and mostly methodologically sound paper (see bellow).

There is one important thing to resolve. The authors need to explain why the geospatial analysis was pertinent. Sure, it produced great maps, but the inferences that made it to the Discussion and Conclusion sections could have been made with simple aggregation instead of the employed GIS methods. If this is not the case, the authors need to explain why their approach was used and discuss that in the Discussion section.

The paper should undergo professional proofreading as some formulations were strange.

There were some minor mistakes and suggestions, which were marked and commented on in the attached PDF.

Reviewer #4: I have reviewed the manuscript and it brings a perspective of spatial and multilevel analysis to understanding the variations in under-five mortality in Ethiopia. The multilevel analysis introduces community level variables not examined before enabling the study to consider factors beyond individual ones. The study uses more recent data. The method of data analysis, that is, multivariate multilevel logistic regression is fine but its limitation is that it does not handle the aspect of censored data well. In performing the analysis, it is not clear in the manuscript if the data were weighted or not. The data source section (page 4, lines 91-108), this could have been summarized further and high light the information collected by the 2019 EMDHS relevant to study. On study variables (page 5, lines 112-122), the presentation of the variables could have been efficiently done in a table with definitions and categorization. More especially for community level variables. There are too many variables considered at individual level. Is it possible to reduce them to relevant ones only based on reviewed literature? For spatial analysis (page 5, lines 124-125), state what was actually done. Highlight the benefits and limitations of using multilevel analysis. In the results section (pages 8-9), is possible to use 1 decimal place for the percentages in the text. Also, check the table titles and ensure they are named appropriately. For multilevel analysis results, the intra-cluster correlation (ICC) is not shown in the table. A theoretical framework work should have been considered by the study. In the discussion section of the manuscript, emphasis should have been on discussing community level factors and the geographical variation in under-five mortality observed. The manuscript should have discussed if the under-five mortality rates in the 2019 EMDHS were reproduced for bench marking prior to deriving its own. The conclusion and recommendations of the manuscript should have been based on the multilevel analysis (community level variables) results in addition to the spatial variation findings on under-five mortality.

Reviewer #5: This paper is well motivated and carries out a series of multivariate and spatial analysis on Ethiopian data on under 5 child mortality. The survey data appears to be of excellent quality from a rigorous sampling design. The spatial analysis is a strong point and allows subregional rates to be detected that are higher or lower than the background. Spatial scan statistics provide a promising approach in this case and are good complements to the other approaches. The authors should properly cite the SaTScan software, along with the author M.Kulldforf (see the user manual). I would encourage citing Stata properly as well (note spelling and case).

In my experience, SaTScan runs need quite a bit of tinkering of the parameter file to get good hot spot representation (i.e., like that in the spatial EB approach) . I think the cluster detection can be improved by selecting a smaller window size or radius in order to detect smaller secondary clusters other than the primary two presented here. Otherwise, this is an informative paper on an important topic. Table 4 could use better formatting. Please consider using coefficient plots. Additional proofing of the manuscript is also recommended to improve clarity.

Reviewer #6: This is a thoughtful and interesting analysis of under five mortality in Ethiopia in the years preceding the current crisis and civil war. The analysis focusses on the major individual and area level variables that have been identified in the literature and the results are largely in line with previous findings. While my overall evaluation is positive there are a number of issues that need to be addressed before publication:

1. The style of writing is extremely telegraphic, to the point that at times the meaning is not strictly clear. For instance, your community level variables refer to place of delivery; toilet use; wealth and media exposure. I presume these are based on majority characteristics from your individual level analysis, but this should be made clearer. In particular, I see no reference to media exposure at the individual level, so what does this variable refer to?

2. A major contribution of this paper is the difference between the different areas of Ethiopia and the relation to community characteristics. It would be important, for non-Ethiopian readers, to provide a bit more background on the different regions, socially and historically. A comparison based on the community characteristics you describe would be particularly useful. You should also provide references for the spatial analysis you present (Moran’s I and the spatial interpolation).

3. Your choice of model (multilevel binomial) is appropriate but the presentation is a bit overloaded. Many of the variables appear to be non-significant in the multivariate model, even if they were significant at the bivariate level. This may be due to collinearity between the variables. Removing the non-significant variables would bring out better the major covariates related to U5M. It is also important to indicate that the coefficients are relative risks (eb). There is no need for both log-likelihood and deviance values (as you need, D = -2LL) but you should indicate the degrees of freedom and the variance of the random effects (clusters) as well as the number of individual cases (level 1) and clusters (level 2).

4. Your discussion focusses on the major factors reducing under 5 mortality, but you do not discuss differences between the models. For instance, use of solid fuel increases U5M in model 2 and reduces it in model 4; Amhara has higher mortality in model 3, lower mortality in model 4. In both cases, the coefficients are non significant (within the CI) but the reverse should still be mentioned (also see previous comment on removing non-significant comparisons).

5. Some of the comments in your discussion appear to be inconsistent. For instance, larger families have lower mortality, but so do higher birth orders. You mention mother’s age at first birth but not at current birth, which may be important in explaining this paradox. You rightly identify risk factors subject to intervention, such as short breast feeding or home delivery, but I fail to see how twin births fall into this category, except in the sense that greater attention should be given to mothers with multiple births.

In sum, this is a valuable contribution which I hope to see published, and my comments are intended to strengthen what is, at heart, a sound analysis based on a rich and valuable dataset.

Reviewer #7: The paper is sound and generallly well-written. The topic is very relevant and the analyses performed compatible and consistent with the research question. I have some minor remarks that I consider should be addressed before approving for publication:

Overall, please explain a bit further the difference between the mini DHS survey and the main DHS survey. Why do you think this new data is helpful to track the trends in U5M. The authors extensively explain the sampling design, but do not explain the difference between the mini and the main DHS survey, which would be helpful in interpreting the results. Is the mini DHS survey comparable to the main DHS survey? Can we use this to compare them in time?

In addition, it would be important to know whether any data quality assessment of deaths was performed before the regression analysis. Would any bias be added to the study due to misreporting of deaths in the survey or issues related to coverage areas? Did the authors correct the registry of deaths in any way or were deaths under 5 taken as it is from the survey? How do the authors feel about this data and do they trust the information?

In addition, some minor remarks/suggestions in the writing part that felt confusing:

lines 82- 84: the sentence is confusing. I would add here a period. and Say: "Though U5M has been declining, it is still high and is taking the lives of many children. Despite this fact, no research on U5M has been done with recent data available from the Ethiopian mini ... "

lines 85-87: similarly, the sentence is confusing. I would suggest: "...to show the current burden of U5M that is crucial for better planning different policies and interventions for U5M prevention. This allows for an efficient allocation of scarce resources according to spatial..."

lines 188: suggestion to make the sentence clearer: "Sociodemographic background" and not "background socio-demographic"

line 195: correct to : "are of poor economic status"

Reviewer #8: This paper aims to examine geographic variation and factors attributable to Under-Five Mortality (UFM) in Ethiopia. Recognizing as an important marker for health equity and access, UFM is considered the best proxy measure or indicator for socioeconomic development. In addition, child mortality rate is also a useful marker of overall development and a Millennium Development Goal (MDG) indicator and its importance has been further emphasized in an ambitious target under the Sustainable Development Goals. Generally, literature search reveals numerous or countless studies on factors determining UFM conducted in the Sub-Saharan African countries, however, spatial distributions and geographic variations are less investigated compared to the UFM factors.

The data sources used, statistical tests and analysis performed systematically support, but NOT to the full extent, in answering the research questions and its intended aims or objectives. Results and conclusion made in relation to research aims and objectives or intentions are supported by data and analysis, in general. However, there needs further refinement and improvement for author(s) to take into the consideration to make this paper more added-value to the knowledge and literature.

I foresee two major drawbacks of this study, which are described below:

1. The Primary Investigator should not limit the independent variables to individual and community level predictors. It is suggested to take into the consideration of inclusion of health interventions, such as malaria, sanitation and hygiene (WASH), reproductive health (RH), vaccinations, micronutrient supplementation and treatments. Most studies have revealed that there are significant association between these health interventions and reduction of UFM. For example, BCG, OPV, Measles, TT, etc. vaccinations have significantly contributed to drastic reduction in deaths of children in most developing countries. Furthermore, measure DHS captures or collects information on health interventions; and it's worthwhile to examine its association with UFM in Ethiopia.

2. Since the other known predictors of child morality are generally attributable to food security and accessibility in conflict areas thus, it is recommended to discuss how such conflicts or disruptions would influence or alter the geographic variations of UFM in Ethiopia? Other risk factors to take into considerations are: indoor air pollution (determined by type of cooking fuel used in household), nutrition, access to basic health services (ANC, PNC, FP, RH, etc.), poverty status by regions, fertility rates, among others.

6. PLOS authors have the option to publish the peer review history of their article (what does this mean?). If published, this will include your full peer review and any attached files.

Reviewer #1: No

Reviewer #2: No

Reviewer #3: No

Reviewer #4: No

Reviewer #5: No

Reviewer #6: **Yes: **Jon Anson

Reviewer #7: No

Reviewer #8: No

---

## [Author Response · Author response to Decision Letter 0]

1 Jun 2022

PLOS ONE 

Point by point response for editors/reviewers’ comments 

The manuscript title “Geographic variation and factors associated with under-five mortality in 2 Ethiopia. A spatial and multilevel analysis of Ethiopian Mini Demographic and 3 Health Surveys 2019”

Manuscript number: PONE-D-21-20589

Dear editor/reviewer,

I would like to thank you for these constructive, building, and improvable comments on this manuscript that would improve the substance and content of the manuscript. We considered each comment and clarification question of reviewers on the manuscript thoroughly. My point-by-point responses for each comment and question are described in detail on the following pages. Further, the details of changes were shown by track changes in the supplementary document attached.

Reviewer comment and authors response

Reviewer #1: 

General comments:

Reviewer's comment: - Overall authors have explained regional variation in under five mortality and the key predictors of U5M in Ethiopia. I have few comments to improve this paper further.

Reviewer's comment:

1. Authors have used random intercept model for binary multilevel logistic regression. Authors can also do some more testing of their model by including random slopes for their significant predictors in this model to check if the effect of their key predictors remains the same for each cluster or is it different. If they don't find any major difference by including the random slopes, then they can provide the current results as it is.

Authors response: thanks for the detailed comment we have tested using the random slop in checking its effect and didn’t find noticeable difference from the random intercept model.

Reviewer's comment:

2. Authors have used different household condition indicators such as toilet clean fuel main roof etc. than wealth index which captures the economic condition of the household better in the analysis. Is there any specific reason to adapt this strategy?

Authors response: Thank you very much for your comment. We have considered using the wealth index as an economic condition of the household in our analysis, and the wealth index variable failed to enter the multivariant multilevel mixed model since the significant level was high during the univariable multilevel mixed model.

Reviewer's comment:

3. Authors should also provide the residuals plots for their final multilevel model in the appendix.

Authors response: thanks again for the detailed comment, we have provided the residual plots for the final model in figure 6 and on page 10, line 248-249, kindly refer to the revised manuscript.

Reviewer's comment:

4. Authors have shown that there is a significant large cluster of population where U5MR is significantly higher than other areas in Figure 4. Authors can include a table with basic socio demographic profile of this high cluster region. This can be helpful to understand which population mostly suffers with high U5M in that region so that specific interventions and programmes can be targeted for those subgroups of population.

Author’s response: Thank you very much for you very constructive comment. In the descriptive statistics section example in table one already reported. We believe that no need to include the specific socio-demographic characteristics of the hot sport areas. If there any help for us we will do it in next round revision. 

Reviewer's comment:

5. Authors should include a short paragraph in the discussion on the implications of their findings in context of the SDG goals.

Authors response: Thank you very much for your comment. We have accepted the comment and kindly refer to the revised manuscript specified on page 10, line 258 up to 260. 

Reviewer's comment:

6. Please see the flow of your texts in the discussion once more and edit the sentences wherever required for more clarity.

Authors response: Thank you very much for your comment. We have accepted the comment and kindly refer to the description part of the revised manuscript.

Reviewer #2: 

General comments:

Reviewer's comment: - This publication is of high importance because it attempts to present sophisticated analyses of a legitimate population-based survey data to highlight key risk factors for under five mortalities in Ethiopia. However, there are two problems with the manuscript: the first is that it is poorly written in English, which makes understanding of all processed hard; secondly, there are some explanations about the statistics used that (partly due to the poor English) are difficult to understand. Double check both the English and the statistical methods used (binary, bivariable? multi-level logistic regression?).

Reviewer response: thank you for your comment. We accept and kindly find the revised manuscript. When we say binary, we want to explain that our outcome variable has only the possibility of failure or success, and also with regards to bivariable, we wanted to illustrate that we first run a multilevel logistic regression model consisting of the outcome variable and one predictor variable for each predictor variables then choosing those predictors that have passed our significant level (less than the P-value of 0.2) to the multivariable multilevel logistic regression model. We used multilevel logistic regression due to the hierarchical nature of the data.

Reviewer #3: 

General comments:

Reviewer's comment: - This is an engaging, insightful paper and mostly methodologically sound paper.

There is one important thing to resolve. The authors need to explain why the geospatial analysis was pertinent. Sure, it produced great maps, but the inferences that made it to the Discussion and Conclusion sections could have been made with simple aggregation instead of the employed GIS methods. If this is not the case, the authors need to explain why their approach was used and discuss that in the Discussion section.

Authors response: thank you for your detailed comment, we accepted and the main reason we employed GIS is for cost minimization and to help the policymakers to focus on the arias noted by GIS and saTscan, we have also edited in the discussion section specifically on page 11, line 300-302 in the revised manuscript.

Reviewer's comment:

The paper should undergo professional proofreading as some formulations were strange.

There were some minor mistakes and suggestions, which were marked and commented on in the attached PDF.

Authors response: thank you for your comments. We accept and kindly find the revised manuscript. 

Reviewer #4

Reviewer's comment: - I have reviewed the manuscript and it brings a perspective of spatial and multilevel analysis to understanding the variations in under-five mortality in Ethiopia. The multilevel analysis introduces community level variables not examined before enabling the study to consider factors beyond individual ones. 

The study uses more recent data. The method of data analysis, that is, multivariate multilevel logistic regression is fine but its limitation is that it does not handle the aspect of censored data well. 

Authors response: thanks for the comment, our data is cross-sectional hierarchical data with a binary outcome and for this type of data multilevel logistic regression is the best method of data analysis.

Reviewer's comment:

In performing the analysis, it is not clear in the manuscript if the data were weighted or not. 

Authors response: thanks for the comment we have accepted and revised it kindly find on page 6 line 166 up to line 167 in the revised manuscript.

Reviewer's comment: 

The data source section (page 4, lines 91-108), could have been summarized further and high light the information collected by the 2019 EMDHS relevant to study. 

Authors response: thanks for the comment we accepted and kindly refer to the revised manuscript specifically (page 4, lines 91- 97).

Reviewer's comment: 

On study variables (page 5, lines 112-122), the presentation of the variables could have been efficiently done in a table with definitions and categorization. More especially for community level variables. 

Authors response: thank you for your comment, we have considered using the table to illustrate the study variables and since the variables are presented in table 1 up to 3 additional tables, we feared that an additional table might be a bit redundant.

Reviewer's comment:

There are too many variables considered at individual level. Is it possible to reduce them to relevant ones only based on reviewed literature? 

Authors response: thanks for your detailed comment, all of our variables are from literature reviews and we have reduced many not significant variables during the bivariable multilevel logistic regression.

Reviewer's comment: 

For spatial analysis (page 5, lines 124-125), state what was actually done. Highlight the benefits and limitations of using multilevel analysis. 

Authors response: thank you for the detailed comment the spatial analysis that was employed are reported in detail on page 5-6, line 127-145.

Reviewer's comment: 

In the results section (pages 8-9), is possible to use 1 decimal place for the percentages in the text. Also, check the table titles and ensure they are named appropriately. 

Authors response: thanks for the detailed comment, we have considered using one decimal place and since we wanted to be precise as possible that is why we used two decimal places. 

Reviewer's comment: 

For multilevel analysis results, the intra-cluster correlation (ICC) is not shown in the table.

Authors response: thanks for the detailed comment, we accept and kindly find the revised table on page 24 in the revised manuscript.

Reviewer's comment: 

A theoretical framework work should have been considered by the study.

Authors response: thank you for the detailed comment, we have provided the theoretical framework kindly find the revised manuscript specifically in figure 1.

Reviewer's comment:

In the discussion section of the manuscript, emphasis should have been on discussing community level factors and the geographical variation in under-five mortality observed. 

Authors' response: thanks for the detailed comment, we only discussed the community factors that are significantly associated with under-five mortality, and most of the community-level factors are not significantly associated with under-five mortality.

Reviewer's comment: 

The manuscript should have discussed if the under-five mortality rates in the 2019 EMDHS were reproduced for benchmarking prior to deriving its own. 

Authors response: thanks for your comment we have discussed benchmarking prior to EDHS, kindly find it on page 10 between the lines 258-261.

Reviewer's comment: 

The conclusion and recommendations of the manuscript should have been based on the multilevel analysis (community level variables) results in addition to the spatial variation findings on under-five mortality.

Authors' response: thanks for the detailed comment, most of the community-level variables were not significantly associated with under-five mortality that is why we didn’t give focused on the community-level variables in the conclusion.

Reviewer #5 

Reviewer's comment: This paper is well motivated and carries out a series of multivariate and spatial analysis on Ethiopian data on under 5 child mortality. The survey data appears to be of excellent quality from a rigorous sampling design. The spatial analysis is a strong point and allows subregional rates to be detected that are higher or lower than the background. Spatial scan statistics provide a promising approach in this case and are good complements to the other approaches.

The authors should properly cite the SaTScan software, along with the author M.Kulldforf (see the user manual). I would encourage citing Stata properly as well (note spelling and case).

Authors response: thank you for your detailed comment, we accepted kindly refer to the revised manuscript specifically page 6, lines 139 and 143.

Reviewer's comment:

In my experience, SaTScan runs need quite a bit of tinkering of the parameter file to get good hot spot representation (i.e., like that in the spatial EB approach) . I think the cluster detection can be improved by selecting a smaller window size or radius in order to detect smaller secondary clusters other than the primary two presented here. 

Authors response: thanks for the detailed comment, we agree that the cluster can be improved by selecting a smaller window size to detect smaller secondary clusters but we aim to identify the primary ones that is why we used the default maximum spatial cluster size of less than 50% of the population, which allows both small and large clusters to be detected.

Reviewer's comment:

Otherwise, this is an informative paper on an important topic. Table 4 could use better formatting. Please consider using coefficient plots. Additional proofing of the manuscript is also recommended to improve clarity.

Authors response: thank you for your detailed comment, we have improved the clarity kindly refer to the revised manuscript. It known that it possible to report using coefficient plot but we believe table four format appropriate in this regard. 

Reviewer #6

Reviewer's comment: 

This is a thoughtful and interesting analysis of under-five mortality in Ethiopia in the years preceding the current crisis and civil war. The analysis focusses on the major individual and area level variables that have been identified in the literature and the results are largely in line with previous findings. While my overall evaluation is positive there are a number of issues that need to be addressed before publication:

1. The style of writing is extremely telegraphic, to the point that at times the meaning is not strictly clear. For instance, your community level variables refer to place of delivery; toilet use; wealth and media exposure. I presume these are based on majority characteristics from your individual level analysis, but this should be made clearer. In particular, I see no reference to media exposure at the individual level, so what does this variable refer to?

Authors response: thank you for your detailed comment, we accepted and kindly refer to the revised manuscript specifically page 5, line 115-123.

Reviewer's comment:

2. A major contribution of this paper is the difference between the different areas of Ethiopia and the relation to community characteristics. It would be important, for non-Ethiopian readers, to provide a bit more background on the different regions, socially and historically. A comparison based on the community characteristics you describe would be particularly useful. You should also provide references for the spatial analysis you present (Moran’s I and the spatial interpolation).

Authors response: thank you for your comment, we didn’t find any significant community characteristics that’s why we didn’t compare based on the community characteristics. Also, we accepted and kindly find the revised manuscript for the edited reference on pages 5, line 130, and 135 respectively.

Reviewer's comment:

3. Your choice of model (multilevel binomial) is appropriate but the presentation is a bit overloaded. Many of the variables appear to be non-significant in the multivariate model, even if they were significant at the bivariate level. This may be due to collinearity between the variables. Removing the non-significant variables would bring out better the major covariates related to U5M. 

Authors response: thanks for the detailed comment, we have checked for collinearity and there was no noticeable collinearity between variables.

Reviewer's comment:

It is also important to indicate that the coefficients are relative risks (eb). There is no need for both log-likelihood and deviance values (as you need, D = -2LL) but you should indicate the degrees of freedom and the variance of the random effects (clusters) as well as the number of individual cases (level 1) and clusters (level 2).

Authors response: thanks for the detailed comment, we accept we have edited out the deviance from the table kindly refer to the revised table 4 in the revised manuscript. As for the number of individual cases and clusters we thought it would be repetition since it has been mentioned in the data source page 4, lines 101 and 105, and also on the result section on page 8, line 194.

Reviewer's comment:

4. Your discussion focusses on the major factors reducing under 5 mortalities, but you do not discuss differences between the models. For instance, use of solid fuel increases U5M in model 2 and reduces it in model 4; Amhara has higher mortality in model 3, lower mortality in model 4. In both cases, the coefficients are non-significant (within the CI) but the reverse should still be mentioned (also see previous comment on removing non-significant comparisons).

Authors response: thanks for the detailed comment. Indeed it is advantageous to discuss the differences between the models, we feer that the paper might loos its pertinence.

Reviewer's comment:

5. Some of the comments in your discussion appear to be inconsistent. For instance, larger families have lower mortality, but so do higher birth orders. You mention mother’s age at first birth but not at current birth, which may be important in explaining this paradox. You rightly identify risk factors subject to intervention, such as short breast feeding or home delivery, but I fail to see how twin births fall into this category, except in the sense that greater attention should be given to mothers with multiple births.

Authors response: thanks for the comment, Since also multiple births are prone to develop short to long-term complications kindly refer to the citation (32). We accept and we edited in multiple births and compared it with singleton birth kindly refer to the revised manuscript specifically on page 12, line 313. 

Reviewer's comment:

In sum, this is a valuable contribution which I hope to see published, and my comments are intended to strengthen what is, at heart, a sound analysis based on a rich and valuable dataset.

Reviewer #7

Reviewer's comment: 

The paper is sound and generally well-written. The topic is very relevant and the analyses performed compatible and consistent with the research question. I have some minor remarks that I consider should be addressed before approving for publication:

Overall, please explain a bit further the difference between the mini-DHS survey and the main DHS survey. Why do you think this new data is helpful to track the trends in U5M? The authors extensively explain the sampling design, but do not explain the difference between the mini and the main DHS survey, which would be helpful in interpreting the results. Is the mini-DHS survey comparable to the main DHS survey? Can we use this to compare them in time?

Authors response: thanks for the detailed comment. Indeed it is advantageous to discuss the differences between the mini-DHS and main DHS, we feer that the paper might loos its pertinence.

Reviewer's comment: 

In addition, it would be important to know whether any data quality assessment of deaths was performed before the regression analysis. Would any bias be added to the study due to misreporting of deaths in the survey or issues related to coverage areas? Did the authors correct the registry of deaths in any way or were deaths under 5 taken as it is from the survey? How do the authors feel about this data and do they trust the information?

Authors response: Thank you very much for your ideal comment. Frankly speaking, we authors had no any room to check death registry. Since is secondary data and there is no strong death registry in the country it is difficult to check death by any other means. Before the data is collected training was given for data collectors this may be one way data quality assurance. The second it large survey and conducted carefully with expertise and representative of the country under-five mortality. The information is trustful. 

Reviewer's comment:

In addition, some minor remarks/suggestions in the writing part that felt confusing:

lines 82- 84: the sentence is confusing. I would add here a period. and Say: "Though U5M has been declining, it is still high and is taking the lives of many children. Despite this fact, no research on U5M has been done with recent data available from the Ethiopian mini ... "

Authors response: thank you for your detailed comment, we accepted and kindly find the edited statements on pages 3 and 4, lines 81 – 84.

Reviewer's comment: 

lines 85-87: similarly, the sentence is confusing. I would suggest: "...to show the current burden of U5M that is crucial for better planning different policies and interventions for U5M prevention. This allows for an efficient allocation of scarce resources according to spatial..."

Authors response: thank you for your detailed comment, we accepted and kindly find the edited statements on page 4 lines 85 – 88.

Reviewer's comment: 

lines 188: suggestion to make the sentence clearer: "Sociodemographic background" and not "background socio-demographic"

Authors response: thank you for your detailed comment, we accepted and kindly find the edited statements on page 8, line 193.

Reviewer's comment:

line 195: correct to: "are of poor economic status"

Authors response: thank you for your detailed comment, we accepted and kindly find the edited statements on page 8, line 199.

Reviewer #8

Reviewer's comment:

This paper aims to examine geographic variation and factors attributable to Under-Five Mortality (UFM) in Ethiopia. Recognizing as an important marker for health equity and access, UFM is considered the best proxy measure or indicator for socioeconomic development. In addition, child mortality rate is also a useful marker of overall development and a Millennium Development Goal (MDG) indicator and its importance has been further emphasized in an ambitious target under the Sustainable Development Goals. Generally, literature search reveals numerous or countless studies on factors determining UFM conducted in the Sub-Saharan African countries, however, spatial distributions and geographic variations are less investigated compared to the UFM factors.

The data sources used, statistical tests and analysis performed systematically support, but NOT to the full extent, in answering the research questions and its intended aims or objectives. Results and conclusion made in relation to research aims and objectives or intentions are supported by data and analysis, in general. However, there needs further refinement and improvement for author(s) to take into the consideration to make this paper more added-value to the knowledge and literature.

I foresee two major drawbacks of this study, which are described below:

Reviewer's comment:

1. The Primary Investigator should not limit the independent variables to individual and community level predictors. It is suggested to take into the consideration of inclusion of health interventions, such as malaria, sanitation and hygiene (WASH), reproductive health (RH), vaccinations, micronutrient supplementation and treatments. Most studies have revealed that there are significant association between these health interventions and reduction of UFM. For example, BCG, OPV, Measles, TT, etc. vaccinations have significantly contributed to drastic reduction in deaths of children in most developing countries. Furthermore, measure DHS captures or collects information on health interventions; and it's worthwhile to examine its association with UFM in Ethiopia.

Authors response: thank you for the detailed comments, we have considered using the above-listed variables, and some we used like WASH variables (time to water), and contraceptive utilization, but still some variables like (water source and toilet type) were not able to pass the bivariable multilevel logistic regression and some variables like (BCG, OPV, measles, and TT) were no available in the EMDHS data.

Reviewer's comment:

2. Since the other known predictors of child morality are generally attributable to food security and accessibility in conflict areas thus, it is recommended to discuss how such conflicts or disruptions would influence or alter the geographic variations of UFM in Ethiopia? Other risk factors to take into considerations are: indoor air pollution (determined by type of cooking fuel used in household), nutrition, access to basic health services (ANC, PNC, FP, RH, etc.), poverty status by regions, fertility rates, among others.

Authors response: thank you for the detailed comments, the EMDHS data doesn’t possess the data that is attributable to food security and whether the EA was a conflict area at the time of collecting the data. additionally, we have considered indoor air pollution and added the variable of cooking fuel to the final models but the results were insignificant and we didn’t want to discuss its results. We have also considered basic health services and most were insignificant during the bivariable multilevel logistic regression model and were not added to the multivariable multilevel logistic regression model.

---

## [Editor Report · Decision Letter 1]

7 Jul 2022

PONE-D-21-20589R1Geographic variation and factors associated with under-five mortality in Ethiopia. A spatial and multilevel analysis of Ethiopian Mini Demographic and Health Surveys 2019.PLOS ONE

Dear Dr. Tadesse Tessema,

Thank you for submitting your manuscript to PLOS ONE. After careful consideration, we feel that it has merit but does not fully meet PLOS ONE’s publication criteria as it currently stands. Therefore, we invite you to submit a revised version of the manuscript that addresses the points raised during the review process.

Please see the very detailed edits we are suggesting in the attached PDF, paying FULL ATTENTION to each and every comment (some of which have been made BEFORE, but have not been addressed). As per our note, make sure your next version is seen by a professional English writer. You will have one last chance to publish the manuscript if it is well written and technically sound. Please contact me if there is any doubts or consultations you may want to make.

We look forward to receiving your revised manuscript.

Kind regards,

Alfredo Luis Fort, M.D., Ph.D.

Guest Editor

PLOS ONE

Journal Requirements:

Additional Editor Comments (if provided):

Several reviewers have seen this document, and have provided lots of editorial and technical feedback. This is an article that has sufficient merit for publication, for global as well as Ethiopian interest. However, despite SEVERAL attempts and specific feedback from reviewers, the authors have failed to submit the manuscript to someone with good English writing skills, so it can be written in "standard English" as is clearly stated in criteria 5 for PLOS ONE publication.

Thus, the document continues to be written in poor English, extremely difficult to read, and still has several editing errors, imprecisions and confusing statements, making it impossible to the average reader to fully grasp the content and comfortably understand all the text, tables and graphs.

We will give you the authors one last chance to revise the manuscript, including giving you a detailed PDF full of comments and editing suggestions) and resubmit, to make a final decision for publication. Again, PLEASE LOOK FOR OR HIRE SOMEONE WITH PROFICIENT ENGLISH WRITING CAPACITY TO REVIEW AND REVISE YOUR MANUSCRIPT ENTIRELY. Resubmit your manuscript only after it has been rewritten ENTIRELY (put full attention to detail!), ensuring everything has been double-checked for technical as well as copy-editing quality. You can submit the typical PDF plus a Word document, in case we may want to make small edits from our end. Thank you.
---

## [Author Response · Author response to Decision Letter 1]

1 Aug 2022

PLOS ONE 

Point by point response for editors/reviewers’ comments 

The manuscript title “Geographic variation and factors associated with under-five mortality in Ethiopia. A spatial and multilevel analysis of Ethiopian Mini Demographic and Health Surveys 2019”

Manuscript number: PONE-D-21-20589R2

Dear editor/reviewer,

I would like to thank you for these constructive, building, and improvable comments on this manuscript that would improve the substance and content of the manuscript. We considered each comment and clarification question of reviewers on the manuscript thoroughly. My point-by-point responses for each comment and question are described in detail on the following pages. Further, the details of changes were shown by track changes in the supplementary document attached.

Reviewer comment and authors response

Reviewer #1: 

General comments:

Reviewer's comment: - THIS IS A GOOD STUDY BUT IS TERRIBLY WRITTEN! THE ONLY WAY IT WILL BE PUBLISHED IS THAT THE AUTHORS LOOK IN DETAIL AT ALL THE IMPRECISIONS AND BADLY WRITTEN CONTENT. ONLY IF THEY RE-WRITE ENTIRELY WITH AN ENGLISH-SPEAKING PERSON (WHOSE FIRST LANGUAGE IS ENGLISH OR IS VERY PROFFICIENT IN ENGLISH), THIS MANUSCRIPT WILL BE PUBLISHED. OTHERWISE, UNFORTUNATELY, IT CANNOT BE PUBLISHED! PLEASE PAY CLOSE ATTENTION TO ALL THE WRITING

Authors response: Thank you very much for you very important comment specially the language of the manuscript and recommendation to give our manuscript for language editor to English speaking person. We accept your comment and gave our manuscript for English speaking person (Elizabeth Korevaar). We acknowledge her support in our revised manuscript. She gave us extensive language edition and we corrected it in the revised version of the manuscript. 

Reviewer's comment: The comment of the reviewer was given on the PDF part of the manuscript with detail comments.

 Authors response: - We accept all comments and corrected it accordingly.

---

## [Editor Report · Decision Letter 2]

2 Sep 2022

PONE-D-21-20589R2Geographic variation and factors associated with under-five mortality in Ethiopia. A spatial and multilevel analysis of Ethiopian Mini Demographic and Health Surveys 2019.PLOS ONE

Dear Dr. Tessema,

Thank you for submitting your manuscript to PLOS ONE. After careful consideration, we feel that it has merit but does not fully meet PLOS ONE’s publication criteria as it currently stands. Therefore, we invite you to submit a revised version of the manuscript that addresses the points raised during the review process. Although the scientific concerns have now been addressed, your submission still requires extensive editing for English grammar and usage. Please note that PLOS ONE does not copyedit accepted manuscripts and that one of our criteria for publication is that articles must be presented in an intelligible fashion and written in clear, correct, and unambiguous English (https://journals.plos.org/plosone/s/criteria-for-publication#loc-5).  A file has been attached containing further edits that should be used as a guide to highlight the changes that need to be made. We suggest you have a fluent English-language speaker thoroughly copyedit your manuscript for language usage, spelling, and grammar. If you do not know anyone who can do this, you may wish to consider employing a professional scientific editing service. While you may approach any qualified individual or any professional scientific editing service of your choice, PLOS has partnered with American Journal Experts (AJE) to provide discounted services to PLOS authors. AJE has extensive experience helping authors meet PLOS guidelines and can provide language editing, translation, manuscript formatting, and figure formatting to ensure your manuscript meets our submission guidelines. If the PLOS editorial team finds any language issues in text that AJE has edited, AJE will re-edit the text for free. To take advantage of this special partnership, use the following link: https://www.aje.com/go/plos/. Please note that PLOS does not financially benefit from this partnership; moreover, having the manuscript copyedited by AJE or any other editing services does not guarantee selection for peer review. Your manuscript has not yet been considered for its scientific merits, and we cannot do so unless the language is significantly improved.

We look forward to receiving your revised manuscript.

Kind regards,

Alfredo Luis Fort, M.D., Ph.D.

Guest Editor

PLOS ONE
---

## [Author Response · Author response to Decision Letter 2]

3 Sep 2022

PLOS ONE 

Point by point response for editors/reviewers’ comments 

The manuscript title “Geographic variation and factors associated with under-five mortality in Ethiopia. A spatial and multilevel analysis of Ethiopian Mini Demographic and Health Surveys 2019”

Manuscript number: PONE-D-21-20589R2

Dear editor,

I would like to thank you for these constructive, building, and improvable comments on this manuscript that would improve the substance and content of the manuscript. We considered each of your recommendation of language edition. Further, the details of changes were shown by track changes in the supplementary document attached.

---

## [Editor Report · Decision Letter 3]

20 Sep 2022

Geographic variation and factors associated with under-five mortality in Ethiopia. A spatial and multilevel analysis of Ethiopian Mini Demographic and Health Surveys 2019.

PONE-D-21-20589R3

Dear Dr. Tadesse Tessema,

We’re pleased to inform you that your manuscript has been judged scientifically suitable for publication and will be formally accepted for publication once it meets all outstanding technical requirements.

Kind regards,

Alfredo Luis Fort, M.D., Ph.D.

Guest Editor

PLOS ONE

Additional Editor Comments (optional):

Final manuscript in PDF sent my author was uploaded from my end. It is OK and should be ready for publication.
---

## [Editor Report · Acceptance letter]

27 Sep 2022

PONE-D-21-20589R3 

Geographic variation and factors associated with under-five mortality in Ethiopia.  A spatial and multilevel analysis of Ethiopian Mini Demographic and Health Survey 2019 

Dear Dr. Tessema:

I'm pleased to inform you that your manuscript has been deemed suitable for publication in PLOS ONE. Congratulations! Your manuscript is now with our production department. 

Kind regards, 

on behalf of

Dr. Alfredo Luis Fort 

Guest Editor

PLOS ONE